# An Empirical Perception of Economic Resilience Responded to the COVID-19 Epidemic Outbreak in Beijing–Tianjin–Hebei Urban Agglomeration, China: Characterization and Interaction

**DOI:** 10.3390/ijerph181910532

**Published:** 2021-10-07

**Authors:** Yaping Zhang, Jianjun Zhang, Ke Wang, Xia Wu

**Affiliations:** 1School of Land Science and Technology, China University of Geosciences (Beijing), Beijing 100083, China; zhangyp1007@163.com (Y.Z.); wangke11260@163.com (K.W.); alisonwuxia@126.com (X.W.); 2Land Consolidation and Rehabilitation Center, Ministry of Natural and Resource, Beijing 100083, China

**Keywords:** COVID-19, economy resilience, economy resistance, economy restoration, China

## Abstract

The COVID-19 has caused a serious impact on the global economy, and all countries are in a predicament of fighting the epidemic and recovering their economies. Aiming to discuss the impact of the COVID-19 on the economic resilience of urban agglomerations, the economic data of each quarter from June 2019 to September 2020 of the Beijing–Tianjin–Hebei Urban Agglomeration are selected, and the economic development index (EDI) is calculated based on the entropy method. Combining the fundamental conditions of urban agglomerations and industrial policies during the COVID-19, urban economic resilience is discussed by the changing trend of the economic development index (EDI) and dividing into resistance and restoration. The results show that: (1) The economic development level of the urban agglomeration has been affected by the epidemic and has changed significantly. The change of endogenous power is the main cause of change; (2) During the outbreak of the COVID-19, the economic resilience of the Beijing–Tianjin–Hebei urban agglomeration shows four different development types: high resistance and restoration, high resistance but low restoration, low resistance but high restoration, low resistance and restoration cities; (3) High resistance but low restoration, low resistance but high restoration, and low resistance and restoration cities influence each other, but the relationship between cities is mainly dependent; (4) The economic restoration within the urban agglomeration forms a synergy, which promotes the economic recovery and development of the urban agglomeration during the recovery period of the COVID-19. Urban agglomerations should enhance the combined effect of resistance and increase the impact of high resistance and restoration cities on surrounding cities in the future.

## 1. Introduction

As a particularly major public health emergency, the COVID-19 became a global pandemic, which made a severe impact on global national economic and social development [1,2]. In China, due to the outbreak of the COVID-19 on the eve of the Spring Festival, the large flow of people returned home before the holiday, which has increased the difficulty of prevention and control of the epidemic [3]. The home quarantine policy had caused an impact on the tourism, transportation, catering, wholesale, retail, entertainment, and other service industries in various regions. The resumption of work after the holiday was also greatly delayed, which caused the national economic development to be obstructed [4]. The urban agglomeration is a unique organizational construction formed with the new urbanization, which plays an important role in promoting the development of various cities [5]. Meanwhile, the urban agglomeration has a high level of economic development, advanced transportation construction, high population density, and close inter-regional relations [6]. Therefore, economic exchanges and frequent population movements have accelerated the spread of the epidemic. While the global epidemic is still in progress [7], analyzing the differences in resilience shown by different cities in responding to the COVID-19 and studying the economic recovery process of urban agglomerations is important for balancing and improving the quality of regional economic development [8].

The outbreak of the COVID-19 in 2020 is a major public health emergency with the fastest spread, the widest range of infections, and the most difficult prevention and control in the world [9]. It has brought a huge impact on the economic development of the world and it caused a huge test of urban economic resilience. Foster [10] and Hill. E [11] think that economic resilience mainly refers to the ability of the economic system to recover from shocks or destruction. Hassink [12] define economic resilience as an important part of urban resilience, is mainly reflected in the ability to continuously adjust its industrial structure and social relations, which can adapt to changes in the external environment and ensure the smooth operation of the economy. The difference in urban economic resilience determines whether a city can recover quickly or fall into a long-term stagnation in the face of a crisis.

## 2. Literature Review

In recent years, natural and artificial disasters have occurred frequently, and urban safety needs in various countries have intensified [13]. Many cities at home and abroad have successively launched resilient city construction. Metropolises in various countries have successively proposed new development concepts to enhance urban resilience in urban planning. Resilience includes not only the ability to resist in the face of crises, but also the restoration to respond to crises. Cellini pointed out that resilience should not only discuss restoration behaviors [14]. Resilience structure can be explained by the adaptation–adaptability relationships between competing, Separated and Reciprocal, etc. [15] This also illustrates the necessity of studying resilience from the perspective of resistance. Research on urban economic resilience has attracted attention after the global financial crisis in 2008. The outbreak of the COVID-19 in 2020 has caused a huge impact on the world economy and a test of urban economic resilience [16,17]. Research on urban economic resilience has received more attention [18]. Di Pietro et al. measured the vulnerability, resistance and recoverability of regions to discuss the resilience and identify key regional features [19]. Islam et al. found that severe adverse impact of the pandemic on global production, employment, and prices [20]. Zhang K. et al. found that the outbreak of the COVID-19 has the greatest impact on the service industry [21]. Many researchers have selected indicators to measure economic resilience. Brown et al. found that industry structures and concentration in particular industries influence economic resilience [22]. Pretorius et al. indicated that economic openness, export market dynamics and sectoral composition may influence economic resilience [23]. Simmie et al. found that regional innovation systems policies can contribute to their economic resilience [24]. Tan J. found that economic development, labor conditions, and the industrial structure had a statistically significant negative effect on economic resilience [25]. Liu et al. pointed out that there are significant differences in economic resilience revealed by multi-dimensional indicators [26]. It provides a reference for constructing an index to measure economic resilience.

The current research scale for analyzing urban economic resilience in the context of the COVID-19 focuses on the world [20], countries [17,21,27], and cities, while there is relatively little research from the perspective of urban agglomerations. Research dimension on resilience needs to further rise from the “urban resilience” to “urban agglomeration resilience” [5]. It is of great significance for improving the emergencies governance level of urban agglomerations to identify the main characteristics of economic resilience of different types of cities. Meanwhile, clarifying the economic resilience of different cities can provide strong support for judgment of the economic situation and urban recovery in the normalized epidemic prevention and control stage [26].

Based on previous research, we discuss the difference of economic resilience by classifying cities from two aspects of resistance and restoration by calculating the economic development index (EDI). The reasons for the differences are analyzed from urban fundamentals, policies and the relationship between urban agglomerations. In addition, suggestions are made for the future development of urban agglomeration under the normalization of epidemic prevention and control. The remainder of this paper is organized as follows. The research framework, methods, and data are introduced in Section 3. Section 4 presents and discusses the resilience results and reasons by modeling analysis and spatial analysis. Based on the analysis results, Section 5 and Section 6 discusses and concludes this study.

## 3. Materials and Methods

### 3.1. Study Area

The Beijing–Tianjin–Hebei urban agglomeration is one of the regions with the most dynamic economy, the highest degree of openness, the strongest innovation capabilities, and the largest population absorption in China [28,29]. It is also the main area of China to participate in regional cooperation in Northeast Asia, which consists of the municipalities of Beijing and Tianjin, along with 11 cities in Hebei Province (Shijiazhuang, Langfang, Qinhuangdao, Hengshui, Chengde, Xingtai, Baoding, Tangshan, Zhangjiakou, Cangzhou, and Handan) (Figure 1). With the continuous advancement of the Beijing–Tianjin–Hebei coordinated development strategy in 2014, the Beijing–Tianjin–Hebei urban agglomeration has achieved industrial agglomeration and in-depth cooperation [30]. Beijing has the ripe high-end manufacturing, while Tianjin and Hebei have strong industrial bases in traditional industries such as heavy and chemical industries, logistics and ports. This urban agglomeration has the national political center, the Chinese–foreign cultural exchange center, and the economic center in the north. To a large extent, it represents the economic development level of China’s well-developed urban agglomeration and plays a pivotal role in China’s economic development.

As the main economic development center, the Beijing–Tianjin–Hebei urban agglomeration is one of the earliest urban agglomerations established in China. With the implementation of the regional coordinated development strategy, Beijing, Tianjin, and Hebei have strengthened their construction of transportation interconnection, comprehensive environmental improvement, and industrial coordination [31]. Meanwhile, the population density of the urban agglomeration is high, the economic industry is developed. The outbreak of the COVID-19 has affected many aspects of the agglomeration such as population flow, industrial division of labor, and urban development in the urban agglomeration, which in turn becomes a threat to urban economic resilience.

### 3.2. Data Sources

The COVID-19 is fast and lasts a long time [32], so this paper selects quarterly economic data before and after the COVID-19(September 2019 to December 2020). The data used for calculating the economic indicators were obtained from the Statistical Yearbook of Beijing, Tianjin and Hebei Province. The local epidemic data come from the real-time updated data of the official websites of the health commissions of the localities, and the policy data come from the policy documents of the official websites of the local governments. For individual missing data, data with the same nature and meaning are used to make up for it.

### 3.3. Methods

#### 3.3.1. Research Framework

The level of urban economic resilience is reflected in the timely and effective use of existing resources to respond to emergencies, and to further repair the dynamic process of economic development in a certain period in the future [12,14,19,33,34]. Research on urban resilience needs to consider both the city’s resistance and the city’s restoration. Firstly, the trend of the economic development index (EDI) is the performance of the city’s economic level under the influence of the epidemic. Therefore, the change in the EDI in the short term caused by the epidemic can be used as dynamic indicators of the city’s economic resilience level. Cities can be divided into different development types based on the results of their resilience. Secondly, economic resilience cannot be measured in a single dimension. A multi-dimensional examination can clarify the nature of economic resilience [26]. Affected by economic resource endowment, infrastructure construction, and the degree of perfection of policies and systems, cities have different performances in adapting to pressures. Due to this, this paper takes the urban fundamentals and industrial policies during the COVID-19 as the primary cause of the differences in urban economic resilience levels under the influence of emergencies. Thirdly, due to the high integration of the cities in the urban agglomeration, the strong economic foundation, and the strong spatial interaction [35], the role of urban agglomerations needs to be considered when discussing the differences in resilience of various cities [36]. So, the spatial correlation characteristics of the economic resilience of various cities is discussed. The research framework of this study is shown in Figure 2.

#### 3.3.2. Selection and Calculation of Urban Fundamentals

Urban economic fundamentals reflect the prosperity of the urban economy [37,38], which are of great significance for measuring economic resilience. What is more, there are differences in medical facilities in cities of different scales, resulting in cities showing different economic resilience in response to the COVID-19. The impact of the COVID-19 on the economy reflects the important role of the city’s economic fundamentals and medical fundamentals in the city’s response to the crisis. Urban economic fundamentals are the support to ensure the survival and sustainable development of the city. Urban medical fundamentals are the most important material bases for the construction of urban. This paper selects economic and medical fundamentals to measure the urban fundamentals, and analyzes the causes of the differences in urban economic resilience.

Indicator system establishment

Considering the accuracy, rationality, and availability of the indicators, this paper selects per capita disposable income, regional GDP, and proportion of tertiary industry in GDP to measure the economic fundamentals of the Beijing–Tianjin–Hebei urban agglomeration [26,39,40,41,42]. The allocation indicators of medical facilities are selected to measure the background of the medical conditions in the urban agglomeration, including the proportion of the area of hospitals, number of hospital beds and number of doctors. Finally, the weight ratios of specific indicators are integrated to determine the differences in urban economic resilience in different urban fundamentals. The specific indicators and their meanings are shown in Table 1.

2.Calculation of indicators

In order to ensure the accuracy of the conclusions, this paper standardizes the indicators.

Positive indicators:(1)$$X_{ij}=\frac{x_{ij}-x_{min}}{x_{max}-x_{min}}$$

Negative indicators:(2)$$X_{ij}=\frac{x_{max}-x_{ij}}{x_{max}-x_{min}}$$
where *x_ij_* refers to the initial value of indicator *i* (*i* = 1, 2, 3, …, m) in the *j* (*j* = 1, 2, 3, …, n) year, and *X_ij_* is the normalized value, *X_ij_* ∈ [0, 1].

#### 3.3.3. Establishment of Economic Development Index (EDI)

Urban economic resilience can be characterized by index changes in a certain period under the influence of emergencies [43,44]. Therefore, the changing trend of the EDI before and after the COVID-19 can reflect the strength of economic resilience. From the perspective of quantity, space and structure, the economic level of a city is affected by the city’s basic industrial structure, geographical location factors, foreign trade as well as the radiation of the surrounding cities (Figure 3). Therefore, based on the development background of the Beijing–Tianjin–Hebei urban agglomeration, this paper selects quarterly data from the basic driving force, external driving force, and endogenous driving force of regional economic development to construct the EDI.

Indicator System Establishment

When calculating the EDI, based on the characteristics of the spatial structure of the Beijing–Tianjin–Hebei urban agglomeration, as well as the connotation of economic basic power (EBP), outside power (EOP), and endogenous power (EEP) [40,45]. GDP growth rate and fixed asset investment growth rate are selected as indicators to measure the EBP; the growth rate of total foreign trade exports and the growth rate of foreign investment absorbed are selected as indicators to measure the EOP; the output value of the secondary industry and the growth rate of total retail sales of social consumer goods are selected to measure the EEP. The specific indicators and their meanings are shown in Table 2.

2.Calculation of Indicator Weight

The entropy evaluation method determines the relative importance of different indicators based on the information entropy implied by the indicator data to calculate indicator weights [46].

Standardization of indicators (Refer to formulas (1) and (2)).
(3)$$f_{ij}=\frac{X_{ij}}{\sum_{j=1}^{n}X_{ij}}$$
(4)$$E_{i}=-\frac{1}{\ln\left(n\right)}\sum_{j=1}^{n}f_{ij}\ln\left(f_{ij}\right)$$
(5)$$w_{i}=\frac{1-E_{i}}{m-\sum_{i=1}^{m}E_{i}}$$
(6)$$\mathrm{EDI}=\sum X_{\mathrm{ij}}\times w_{i}$$
where *f_ij_* is the rate of the *x_ij_* in the sum of the standardized values of the indicator, *E_i_* is the information entropy of *x_ij_*, and *w_i_* is the indicator weight of indicator, *w_i_* ∈ [0, 1], EDI ∈ [0, 1].

#### 3.3.4. The Resilience of Urban Economy

In the part of framework designing, we construct a model named urban economic resilience model in the crisis through economic resistance and restoration (Figure 4). The curve simulates the trend of economic development index. The resistance is *f*(*x*), which is the minimum EDI change rate (*k*_1_) from before the outbreak of the crisis to the study period. The restoration is *g*(*x*), which is the maximum EDI change rate (*k*_2_) from the study period after the crisis broke out.
(7)$$f\left(x\right)=\min(k_{1})=\frac{EDI_{1}-EDI_{min}}{t_{1}-t_{min}}$$
(8)$$g\left(x\right)=\max(k_{2})=\frac{EDI_{2}-EDI_{min}}{t_{2}-t_{min}}$$

This paper uses the rate of change of the EDI to measure the economic resilience of different cities. In order to be comparable, based on the first, middle, and last three time points of the crisis. The lowest critical point of the EDI, the EDI in December 2019, and the December index in the same period of 2020 are selected. This paper calculates the two-point slope of the difference between the data in December 2019 and the critical index point as the city’s resistance to judge the city’s ability to resist the impact of the epidemic, and calculates the two-point slope of the difference between the data in December 2020 and the critical index point as the city’s restoration to judge the city’s ability to recover after the epidemic.
(9)$$r_{1}=f\left(x\right)=\frac{EDI_{1}-EDI_{min}}{t_{1}-t_{min}}$$
(10)$$r_{2}=g\left(x\right)=\frac{EDI_{2}-EDI_{min}}{t_{2}-t_{min}}$$
where *r*_1_ is the city’s resistance. *r*_2_ is the city’s restoration. *EDI*_1_ is the *EDI* in December 2019 and *t*_1_ is December 2019, *EDI*_2_ is the EDI in December 2020 and *t*_2_ is December 2020, *EDI_min_* is the minimum of the EDI and *t_min_* is the time of the lowest critical point.

## 4. Results

### 4.1. Urban Fundamentals

By calculating the economic fundamentals and medical fundamentals of each city in the Beijing–Tianjin–Hebei urban agglomeration, the results are shown in the Figure 5. Beijing and Tianjin show the strongest level of economic and medical fundamentals. Other cities belong to Hebei Province, which economic fundamentals are similar, but the medical fundamentals are quite different. The urban fundamentals of the Beijing–Tianjin–Hebei urban agglomeration show great differences, and there is an imbalance in the allocation of regional resources in the urban agglomeration. Besides, the overall urban medical fundamentals and economic fundamentals show a relatively significant positive correlation. Cities with better economic fundamentals have relatively better medical fundamentals. It shows that as the economic development shifts from high-speed development to high-quality development, the urban medical basic security system is becoming more excellent, turning to the overall high-quality development of the city.

### 4.2. Economic Performance under the Influence of the COVID-19

#### 4.2.1. Economic Performance of the Beijing–Tianjin–Hebei Urban Agglomeration

By calculating the changes in the indicators, and combining local epidemic data from the real-time update data of the official websites of the health commissions of the localities and policies from the policy documents of the official websites of the local governments, we draw the trend of the economic development index before and after the COVID-19 (Figure 6). The EDI of the Beijing–Tianjin–Hebei urban agglomeration has been greatly affected by the COVID-19. With the outbreak stage from December 2019 to March 2020, the EDI showed a significant downward trend. Local governments have actively taken corresponding measures to respond to the COVID-19, and timely promulgated epidemic prevention and control policies to prevent the further spread of the epidemic. The number of newly diagnosed patients began to decline after the peak in March 2020, and the COVID-19 was brought under control. Due to the lag effect in the policy response of some cities, the EDI of the Beijing–Tianjin–Hebei urban agglomeration showed a certain downward trend after March 2020. However, the downward trend has obviously slowed down, and after June it began to show an upward trend, and the rising rate has gradually increased.

After the outbreak of the COVID-19, the main influencing factor that led to a significant downward trend in the EDI was the significant decline in the EEP. Because the impact of the COVID-19 caused the suspension of work and production, which had a larger impact on the production and sales of industrial enterprises, and caused a significant decline in the output value of the secondary industry. Coupled with the suspension of traffic during the epidemic, many laborers were unable to return to work in time, which also greatly affected the resumption of work and production. In addition, the spread of the COVID-19 throughout the country during the Spring Festival has led to a reduction in population flow. Service industries and retail industries such as tourism and catering are also affected to a certain extent.

#### 4.2.2. Economic Performance of Each City in the Beijing–Tianjin–Hebei Urban Agglomeration

By calculating the changes in the indicators, we draw the trend of the economic development index before and after the COVID-19 in different cities (Figure 7). The overall EDI of the cities in the Beijing–Tianjin–Hebei urban agglomeration shows relatively similar trend, but the magnitudes of change are significantly different. Before the outbreak of the COVID-19 (December 2019), the EDI of the urban agglomeration showed a relatively stable upward trend. The coordinated development and cooperation between cities in the urban agglomeration has been further deepened, and the overall competitiveness of the urban agglomeration has been further improved. After the outbreak of the COVID-19 in early 2020, the EDIs of various regions showed downward trends. Until the epidemic was controlled in the second quarter, the EDIs of some cities began to rise. There are differences in the basic level of economic development, urban fundamentals, and industrial policies between cities. Therefore, the impact of the COVID-19 has caused different change ranges in the EDIs. The difference in the degree of decline and the speed of recovery in the EDI of each city reflects the difference in economic resilience in the event of a public health crisis.

### 4.3. Economic Resilience of the Beijing–Tianjin–Hebei Urban Agglomeration

#### 4.3.1. Types of Economic Resilience of the Beijing–Tianjin–Hebei Urban Agglomeration

Based on the definition of economic resilience, this paper summarizes it as resistance and restoration. We divide resilience into four types, including strong cities (SC), turbulent cities (TC), self-supporting cities (SSC) and fragile cities (FC) (Table 3). Using the economic resilience model, this paper calculates the resilience and restoration of the cities in the Beijing–Tianjin–Hebei urban agglomeration (Figure 8).

The Characteristics of Strong Cities (SCs)

SCs can effectively resist the impact of the epidemic, while being able to recover and adjust in time. It is mainly because of the improvement of the coordinated promotion of epidemic prevention and control as well as economic and social development basis. After the outbreak of the COVID-19, those cities took some policies to respond quickly in time, which can ensure the stable operation of the economy. Ensuring the supply of materials during the epidemic, supporting the resumption of work and production of enterprises, increasing fiscal, taxation and financial support, and supporting the development of key industries helped the economy recover quickly. Zhangjiakou, Tianjin, Qinhuangdao, and Hengshui belong to SCs. Tianjin has a strong economic and industrial structure, and the economic fundamentals are relatively stable. Despite the poor urban fundamentals, Zhangjiakou, Qinhuangdao, and Hengshui can respond quickly in the face of crisis, which also reflects the importance of emergency management for urban development.

2.The Characteristics of Turbulent Cities (TCs)

TCs are less affected by the epidemic, and rely on their fundamentals to cope with the spread of the epidemic in the short term. These cities have certain industrial or economic and political advantages, which can withstand a crisis with sufficiently strong fundamentals. However, the development of urban economy is unstable, the transformation and upgrading of urban pillar industries and traditional industries are lacking. The lack of digital technology-supported new industries and new business formats can timely “supplement” economic development. It is difficult for these cities to recover in time from economic turmoil when facing the long-term impact of the crisis.The COVID-19 has continuous impacts on the EDI, resulting in low restoration. Shijiazhuang and Xingtai are TCs.

3.The Characteristics of Self-Supporting Cities (SSCs)

SSCs lacked emergency management capabilities in responding to the epidemic after the outbreak of the COVID-19, resulting in low resistance. The relatively large number of confirmed cases of the epidemic has caused a greater impact on their economy. As time goes by, the strong fundamentals in cities and the implementation of subsequent epidemic prevention and control mechanisms have enabled the epidemic to be effectively controlled. Policies on the resumption of work and production of enterprises, support for small and micro-enterprises, and strengthening of employment security have been introduced to help the resumption of economy. Including important port city Tangshan, the urban fundamentals are developed. Chengde’s urban fundamentals are poor, so it is difficult to fight the epidemic. However, influenced by Tangshan, its economic development gradually recovered.

4.The Characteristics of Fragile Cities (FCs)

FCs have been severely affected by the COVID-19 for a long time, which has a greater impact on the city’s economy. During the epidemic, the suspension of work and production as well as the reduction of population movement caused the economic industry to be stagnant. The epidemic has caused a continuous impact on the urban economy, leading to a slower recovery and development of economic industries. Beijing, Baoding, Langfang, Cangzhou and Handan are FCs. Differences in economic resilience cannot be explained simply by geographical location and political advantage. Baoding, Langfang, Cangzhou, and Handan with poor urban fundamentals are greatly affected by the industrial transfer in Beijing. As the world’s urbanization process, large cities are becoming an important focus of spreading infectious diseases [13]. Beijing is relatively special, the risk of epidemic spread is high, and the prevention and control of the epidemic is difficult. As a result, its EDI declines quickly and rebounds slowly. This also reflects that, compared with the urban fundamentals, the management ability with an urban emergency response is more important in the face of major health emergencies.

We visualize cities with different types of economic resilience (Figure 9). SSCs, TCs, and FCs show significant spatial agglomeration and the centers are Tangshan, Shijiazhuang, and Beijing, respectively, based on their relatively strong fundamentals. Among them, TCs are mainly distributed in the north of the Beijing–Tianjin–Hebei urban agglomeration. As a port city, Tangshan is close to Beijing and Tianjin, with a superior geographical position and a strong economic foundation, which has a positive effect on the economic development of Chengde. SSCs are mainly distributed in the south of the Beijing–Tianjin–Hebei urban agglomeration. Shijiazhuang, the capital of Hebei Province, is one of the growth poles in the Beijing–Tianjin–Hebei region. It has strong urban fundamentals, and formed “the certain spillover effect.” Most of the FCs are located in the central area of the Beijing–Tianjin–Hebei urban agglomeration. With the evacuation of Beijing’s non-capital functions, Baoding and Langfang are mainly affected by Beijing’s industrial transfer. Therefore, under the influence of the epidemic, the resilience of these cities has shown similar characteristics. At present, the cities relation in the urban agglomeration are mainly dependent on attachment, lack of cooperation, and reciprocity.

#### 4.3.2. Response of Economic Resilience of Beijing–Tianjin–Hebei Urban Agglomeration

As urban agglomerations have close economic exchanges and frequent population movements, the economic impact of the COVID-19 on the Beijing–Tianjin–Hebei urban agglomeration is spatially interactive. This paper discusses the spatial correlation characteristics of economic resilience of Beijing–Tianjin–Hebei urban agglomeration. By calculating the global Moran’s I of resilience (Figure 10), the global Moran’s I of urban agglomeration economic resistance is −0.189, and the P-value is 0.32. The significance test is not passed, indicating that the urban agglomeration resistance has not yet formed a synergistic effect when faced the COVID-19. A unified “anti-epidemic” response was not reached. The global Moran’s I of urban agglomeration economic restoration is 0.435, and the P-value is 0.006, passing the significance test of 0.01. The restoration of the Beijing–Tianjin–Hebei urban agglomeration has positive correlation characteristics. During the period of economic recovery, the Beijing–Tianjin–Hebei urban agglomeration established a joint work mechanism for joint prevention and control of the Beijing–Tianjin–Hebei new crown pneumonia epidemic and achieved positive results.

## 5. Discussion

### 5.1. Main Achievements

From the perspective of economic resilience, this paper provides a case and a new analysis perspective for urban agglomerations to respond and govern the public health emergencies in the future. Affected by economic resource endowment, infrastructure construction, and the degree of perfection of policies and systems, cities have different performances in facing crises. This paper discusses the economic resilience of the Beijing–Tianjin–Hebei urban agglomerations under the impact of the COVID-19 from two perspectives of resistance and restoration, as well as summarizes the urban agglomerations into four types. SCs have high urban resilience both resistance and restoration, which can effectively deal with the impact of the epidemic. TCs have shown strong resistance, but their restoration is relatively weak. The epidemic is not good for their long-term development. SSCs have weak resistance and strong restoration. Due to a lack of emergency management capabilities, the epidemic is likely to cause fluctuations in these cities in the short term. FCs have weak resistance and restoration and the epidemic has hit them harder. This classification method is also consistent with the current common classification of regional economic resilience [47].

### 5.2. Limitations and Uncertainties

This paper proposed a method to measure the resilience of urban economy under the impact of public health emergencies by using short-term economic development index, which can better reflect the sudden impact of public health emergencies. Because the outbreak has not yet been fully contained globally, the data currently available cannot be used to discuss the lasting impact of the outbreak. Urban economic resilience is a diverse and complex concept, which is affected by multiple factors. After the data are perfected in the future, indicators can be further added to measure the urban EDI, making the measurement of urban economic resilience more accurate. Regardless, our study at least evaluates resilience differences in cities using existing data under the impact of emergencies, and the research conclusion provides a useful diagnosis and helpful information for urban agglomeration construction.

When studying the influencing factors of EDI, this paper draws a statistical conclusion according to the index weight ratio. From a macro perspective, the economic endogenous power is the main reason for the decline of EDI, and urban agglomeration should focus on protecting endogenous power when dealing with emergencies. Only macro statistical laws are shown when showing the internal factors mechanism of the EDI that is a highly susceptible factor, and lack an analysis of the causal relationship between specific indicators and EDI. In the future, this study will be better supported by statistical conclusions by constructing an analysis model of influencing factors. For example, the main influencing factors of urban economic resilience can be analyzed by constructing a regression model.

The current research takes the Beijing–Tianjin–Hebei urban agglomeration as an example to analyze the differences in urban economic resilience within the urban agglomeration under the influence of the COVID-19, and whether the Beijing–Tianjin–Hebei urban agglomeration has played a synergistic role in the economic resilience of each city. The comparison of different urban agglomerations can clarify the differences in the economic resilience of urban agglomerations, thereby promoting the improvement of urban agglomeration governance capabilities. In the future, we can compare the Beijing–Tianjin–Hebei urban agglomerations with other types of urban agglomerations around the world to analyze the differences in economic resilience under the influence of the COVID-19, especially to discuss economic development differences under the normalization of the epidemic. It is conducive to realizing the scientific layout of urban agglomeration functions, space, transportation, and form, and to deal with the relationship between concentration and decentralization [5].

### 5.3. Implications and Applications

The outbreak of the COVID-19 in early 2020 has had a greater impact on the urban economic development of the Beijing–Tianjin–Hebei urban agglomeration. The COVID-19 epidemic has been brought under control through proactive response measures, and economic activity is slowly recovering. There are certain differences in the change trends of the EDI of each city, which reflect the difference in the level of urban resilience. With the effective control of the COVID-19, the economic level of urban agglomerations will continue to rise. The upward trend mainly includes two situations, which are recovery to the pre-recession rate (b) or to a sustained higher (a)/lower (c) growth rate (Figure 11) (Only simulating the trend of change, the curve will fluctuate with the actual situation). Moreover, the economic resilience of each city is one of the main influencing factors. Meanwhile, it is also affected by the extent and duration of the emergency.

The lack of synergy in the resilience of the Beijing–Tianjin–Hebei urban agglomeration after the outbreak of the COVID-19 reflects the inadequate emergency management capacity of the urban agglomeration as an overall economic system. Meanwhile, within the urban agglomeration, there is a lack of cooperation and reciprocity. The epidemic was finally controlled thanks to the relevant prevention and control policies issued by the local government. Urban agglomerations have strong population and economic agglomeration. Therefore, local government should actively play the joint role of urban agglomerations when responding to the crisis. In the future, urban agglomeration construction should not only focus on restoration improvement, but also further strengthen resistance construction. On the one hand, urban agglomeration should actively promote medical and health cooperation, share medical and health resources, build medical and health informatization, and improve medical service application system. On the other hand, a global crisis such as the COVID-19 pandemic requires a global response, not only on health but also on trade, finance, and macroeconomic policies [48]. To cope with the economic construction under the normal situation of epidemic prevention and control, urban agglomerations need to build a moderately gradient industrial division structure, and further rely on industrial clusters to form synergistically supported industrial clusters, which can help the agglomeration form more cooperative relations.

By classifying cities according to the characteristics of economic resilience, it helps local governments to better formulate relevant economic policies in the future. Different cities in the urban agglomeration have their development characteristics, especially facing crisis. Although SCs have a strong industrial foundation and emergency response capabilities, which can resist the impact of the epidemic, they have not played a spreading role and have no positive impact on the surrounding cities. In the future, those cities should further strengthen their ties with neighboring cities to enhance their joint efforts to cope with the crisis. SSCs have economic and industrial foundations, while the epidemic embodies the lack of urban emergency management capacity. Therefore, an effective public health early warning system should be built in the future, the early warning responsibility mechanism of all levels of departments should be strengthened and the regional economic resistance to pressure. Most of TCs are less affected by the epidemic and can resist the epidemic in the short term. After the epidemic is controlled, TCs are difficult to resume development promptly and lack certain innovation capabilities. It is necessary to build pillar industries in the city, vigorously promote the development of new technologies, promote sustained and stable economic recovery, as well as enhance the development of foreign trade. FCs are supposed to be the focus of the future construction of urban agglomerations. It is necessary to promote the development of new technologies, and promote sustained economic recovery in most of FCs. Besides, those cities should enhance the city’s emergency management capabilities and improve the early warning system.

## 6. Conclusions

Taking the Beijing–Tianjin–Hebei urban agglomeration as an example, this paper combined traditional epidemiological survey data, short-term economic development level data, and urban development background data to calculate economic development index (EDI). Based on the differences of resource endowments, development models, policies, and background conditions of each city, the reasons for differences in urban economic resilience were discussed.

The urban fundamentals of the Beijing–Tianjin–Hebei urban agglomeration have significant internal differences, and there is still a certain gap in the background conditions of Hebei Province with Beijing and Tianjin. In the future, the central city must play a leading role in the coordinated development of the region to achieve the coordination of urban fundamentals of the urban agglomeration background. Based on the changing trend of the economic development index (EDI), we discussed economic resilience in terms of resistance and restoration, and divided cities into four types, including Strong Cities (SCs), Turbulent Cities (TCs), Self-supporting Cities (SSCs), Fragile Cities (FCs). It provides a research idea for studying the economic resilience of different cities in urban agglomerations, and is conducive to making development suggestions tailored to local conditions. Additionally, we found that the central cities of FCs, SSCs and TCs affect surrounding cities and urban agglomeration played a synergistic role in economic recovery after the epidemic was brought under control. In the future, the urban agglomeration needs to play a synergistic role after the outbreak of the crisis to improve urban resilience.

## Figures and Tables

**Figure 1 ijerph-18-10532-f001:**
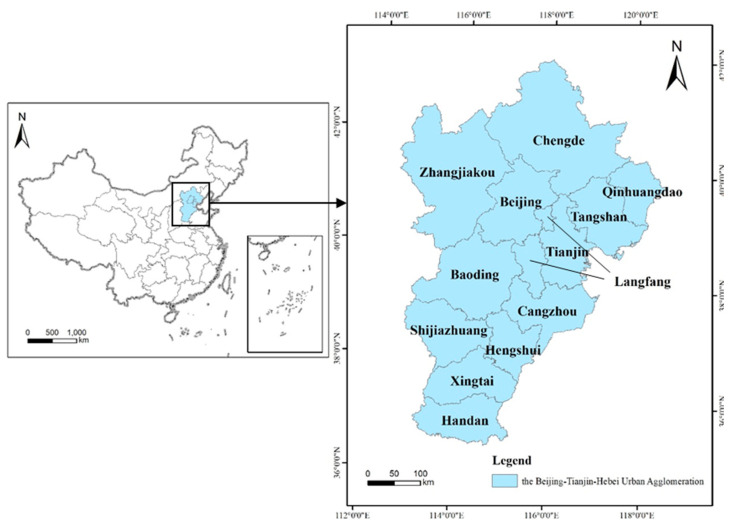
The location and range of the Beijing–Tianjin–Hebei urban agglomeration. Data Source: http://bzdt.ch.mnr.gov.cn/ (accessed on 23 June 2021).

**Figure 2 ijerph-18-10532-f002:**
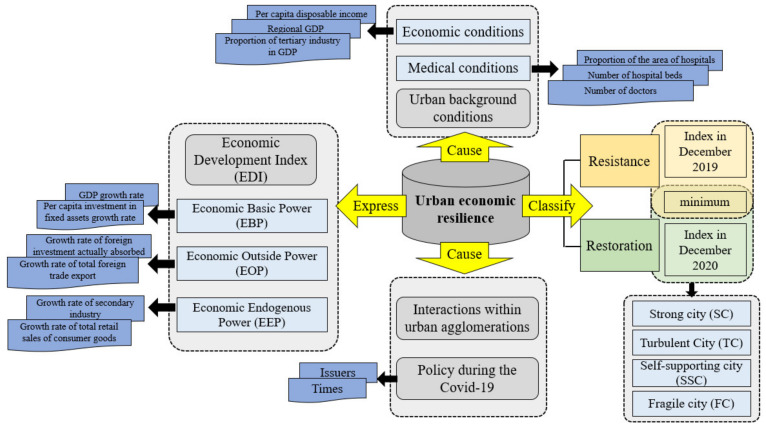
The research framework of this study.

**Figure 3 ijerph-18-10532-f003:**
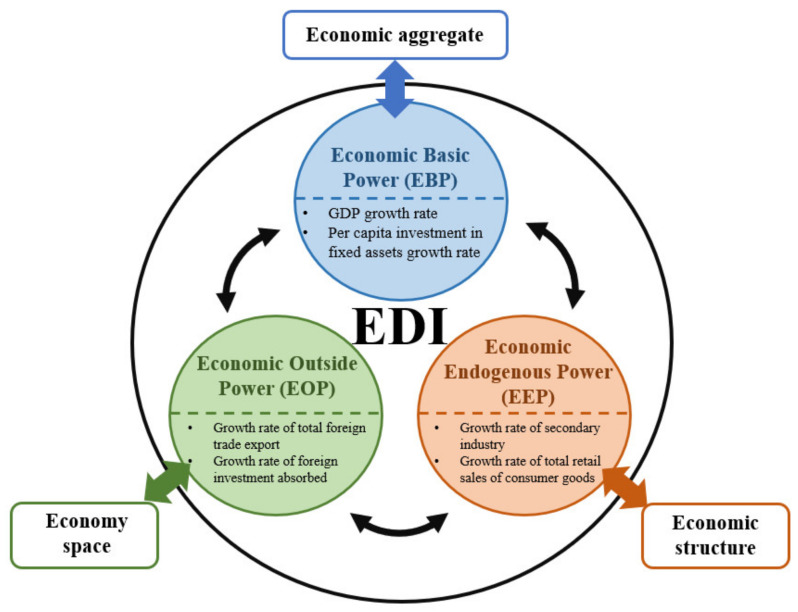
The EDI Frame of Beijing–Tianjin–Hebei Urban Agglomeration.

**Figure 4 ijerph-18-10532-f004:**
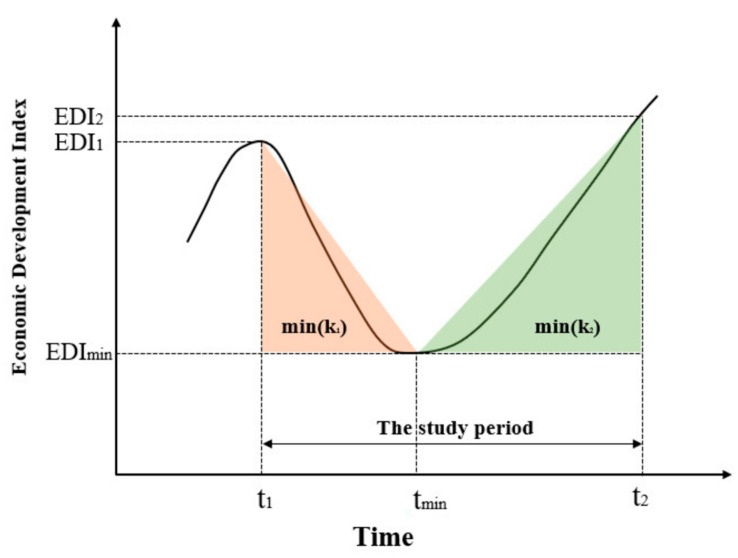
Conceptual Diagram of Economic Resilience Model.

**Figure 5 ijerph-18-10532-f005:**
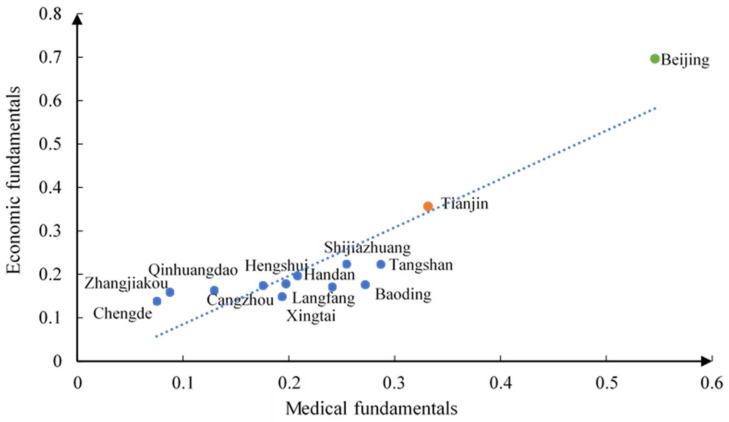
Urban fundamentals of the Beijing–Tianjin–Hebei urban agglomeration.

**Figure 6 ijerph-18-10532-f006:**
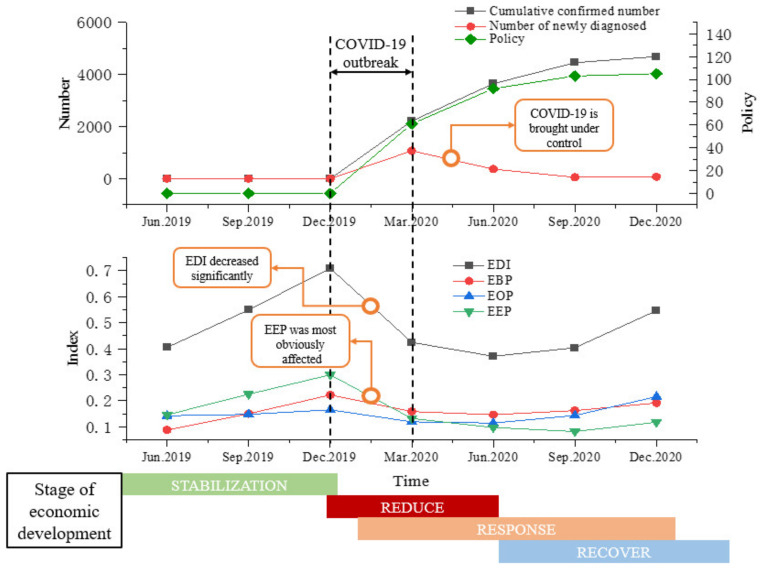
The EDI of the Beijing–Tianjin–Hebei urban agglomeration, the trend of confirmed number and industrial policy before and after the COVID-19.

**Figure 7 ijerph-18-10532-f007:**
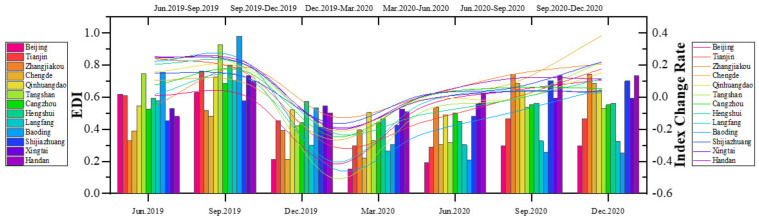
Changes in the EDI of each city in the urban agglomeration before and after the COVID-19.

**Figure 8 ijerph-18-10532-f008:**
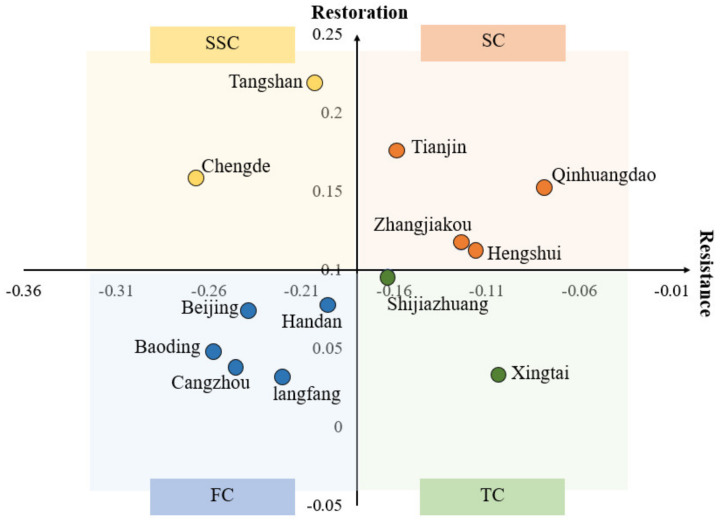
The distribution of economic resilience of the Beijing–Tianjin–Hebei urban agglomeration.

**Figure 9 ijerph-18-10532-f009:**
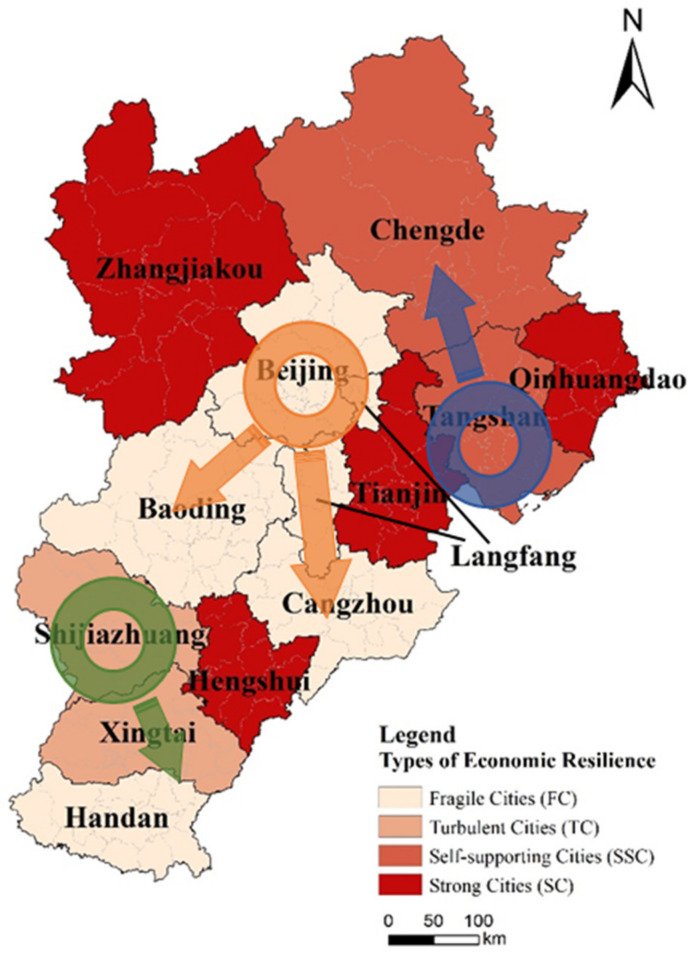
Visualization of economic resilience of Beijing–Tianjin–Hebei urban agglomeration.

**Figure 10 ijerph-18-10532-f010:**
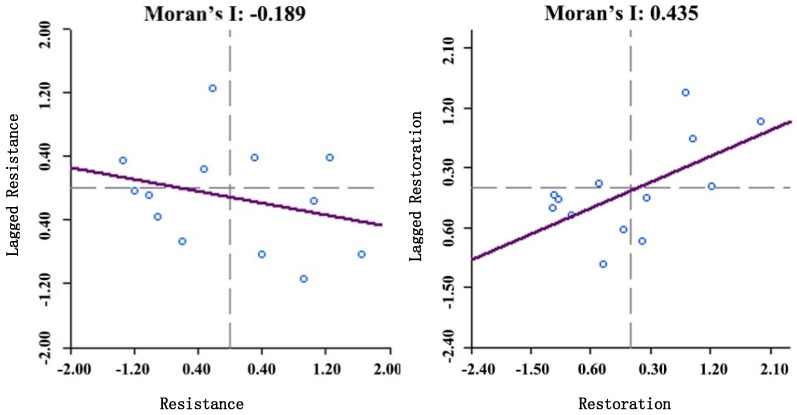
The global Moran’s I of resistance and restoration of the Beijing–Tianjin–Hebei urban agglomeration.

**Figure 11 ijerph-18-10532-f011:**
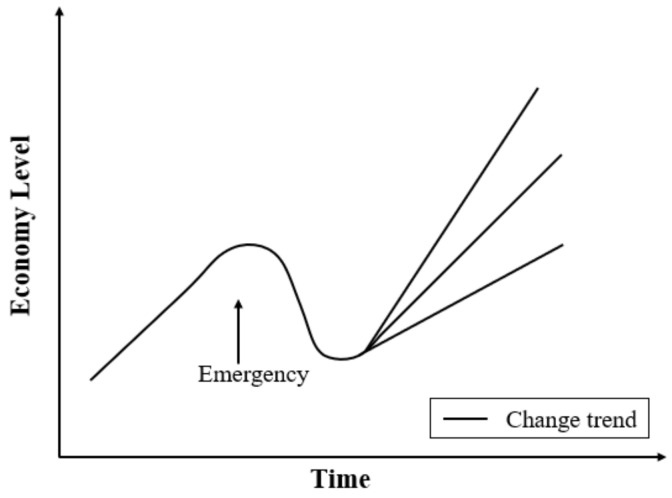
Impacts of an emergency shock on a city’s economy: (a) resumption of a unchangeable rate (pre-recession); (b) resumption of a sustained higher growth rate; (c) resumption of a sustained lower growth rate.

**Table 1 ijerph-18-10532-t001:** The indicator system of the urban fundamentals of the Beijing–Tianjin–Hebei urban agglomeration.

Criterion Layer	Indicator Layer	Indicator Interpretation	Data Source
Economic fundamentals	Per capita disposable income	Regional economic strength and market size, the level of consumer spending in the regional economy and the degree of economic development of the region	Statistical yearbooks and statistical bulletins of Beijing, Tianjin, and Hebei
	Regional GDP		
	Proportion of tertiary industry in GDP		
Medical fundamentals	Proportion of the area of hospitals	The configuration of medical facilities in cities of different sizes	Statistical yearbooks and statistical bulletins of Beijing, Tianjin, and Hebei
	Number of hospital beds		
	Number of doctors		

**Table 2 ijerph-18-10532-t002:** The EDI System of Beijing–Tianjin–Hebei Urban Agglomeration.

Criterion Layer	Indicator Layer	Indicator Interpretation	Data Source
Economic Basic Power (EBP)	GDP growth rate	The economic development status of a region over a period of time	Statistical yearbooks and monthly reports of Beijing, Tianjin, and Hebei
	Per capita investment in fixed assets growth rate		
Economic Outside Power (EOP)	Growth rate of foreign investment absorbed	The level and trend of regional foreign economy	Statistical yearbooks and monthly reports of Beijing, Tianjin, and Hebei
	Growth rate of total foreign trade export		
Economic Endogenous Power (EEP)	Growth rate of secondary industry	The economic development mode of a region and the vitality of regional economic development	Statistical yearbooks and monthly reports of Beijing, Tianjin, and Hebei
	Growth rate of total retail sales of consumer goods		

**Table 3 ijerph-18-10532-t003:** Types of Economic Resilience.

Types of Economic Resilience	Resistance Level	Restoration Level	Resilience Characteristics
Strong Cities (SCs)	high	high	Respond quickly and recover in time
Turbulent Cities (TCs)	high	low	Respond quickly and recover slowly
Self-supporting Cities (SSCs)	low	high	Respond overwhelmingly and recover in time
Fragile Cities (FCs)	low	low	Respond overwhelmingly and recover slowly

## Data Availability

The datasets used in this research are available upon request.

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
