# Peer review of "An Empirical Perception of Economic Resilience Responded to the COVID-19 Epidemic Outbreak in Beijing–Tianjin–Hebei Urban Agglomeration, China: Characterization and Interaction"

_ijerph, 2021, doi:10.3390/ijerph181910532_

Round 1

Reviewer 1 Report

The research topic is interesting. Moreover, the paper in general terms is well written, and has a pertinent topic. 

The authors should consider the following recommendations in order to improve the original manuscript:

  • Introduction and Literature review should be two separate sections, not a common mixed and incoherent section as now. It is more than necessary to include a new section "Literature review". The authors also did not provide sufficient evidence on literature review to support the hypotheses. The Introduction section also includes the Literature review section which is practically non-existent being mentioned only a few bibliographic references quite uncorrelated. Authors should take into consideration much more recent publications in the sphere of discussed subject matter, especially studies conducted during the last 5 years. Please discuss in detailed manner about Severe Acute Respiratory Syndrome Coronavirus 2 (SARS-CoV-2) and its impact on economy. I suggest extending the literature section by including at least the following relevant studies:

- Batool, M., Ghulam, H., Hayat, M.A., Naeem, M.Z., Ejaz, A., Imran, Z.A., Spulbar, C., Birau, R. & Gorun, T.H. (2020) How COVID-19 has shaken the sharing economy? An analysis using Google trends data, Economic Research-Ekonomska Istraživanja, DOI: 10.1080/1331677X.2020.1863830;

- Hayat, M.A.; Ghulam, H.; Batool, M.; Naeem, M.Z.; Ejaz, A.; Spulbar, C.; & Birau, R. (2021) Investigating the Causal Linkages among Inflation, Interest Rate, and Economic Growth in Pakistan under the Influence of COVID-19 Pandemic: A Wavelet Transformation Approach, Journal of Risk and Financial Management, 14(6):277. https://doi.org/10.3390/jrfm14060277.

  • Deepen the description of the limitations of conducted research and indicate the trends for further empirical research.
  • To expand the managerial implications in the article.
  • The sources must be added under each table and figure.
  • The Conclusions section is completely missing from this paper, only “Discussion” section (?!?). Please include a suitable new section entitled “Conclusions” in order to provide all relevant aspects concluded by your research study.
  • Human proofreading, English grammar and spelling correction are also required in order to improve the quality of the manuscript.

Reviewer 2 Report

The work appears well thought out and structured. The topic is relevant in this period. The methodological aspects should be strengthened. In particular I think that the empirical results must be inserted within a robust model. It should be better specified how the results presented were achieved.

 The paper merit publication in the journal and it can be accepted with minor revision

Round 2

Reviewer 1 Report

The original manuscript has been significantly improved. The authors followed the recommendations included in the previous review report so that the quality of their research article has greatly increased. The revised version of the manuscript complies with the Mathematics journal standards. I also appreciate the hard effort of the authors in this regards.